# Helocrenic springs as sources of nutrient rich fine particulate organic matter in small foothill watershed

**Kamila Tichá**[1,2]\*, **Ondřej P. Simon**[1,2], **Jakub Houška**[3], **Lucie Peláková**[2], **Karel Douda**[4]

**1** Department of Special Hydrobiology and Ecology, Branch of Applied Ecology, T.G.M. Water Research Institute, Prague, Czech Republic, **2** Department of Ecology, Faculty of Environmental Sciences, Czech University of Life Sciences Prague, Prague, Czech Republic, **3** Department of Soil Science and Soil Protection, Faculty of Agrobiology, Food and Natural Resources, Czech University of Life Sciences Prague, Prague, Czech Republic, **4** Department of Zoology and Fisheries, Faculty of Agrobiology, Food and Natural Resources, Czech University of Life Sciences Prague, Prague, Czech Republic

\* kamila.ticha@vuv.cz

**Data Availability Statement:** All relevant data are within the manuscript.

**Funding:** KT, OPS, LP: grant nr. MZP 0002071101, Ministry of Environment of the Czech Republic (https://www.mzp.cz/en) KD, JH: grant nr. 13-05872S, Czech Science Foundation (https://gacr.

## Abstract

Despite the large number of studies devoted to organic matter dynamics in fluvial eco-systems, the detrital pathways of spring headwater systems remain neglected. In particular, spring wetlands (helocrenes or seepages) might have considerable influence on downstream headwater stream systems due to the alteration of the nutrient and organic matter content of the water. In this study, we examined fine particulate organic matter (FPOM) drained from helocrenic springs to describe its downstream transport. We studied the quantity, nutrient content and physical components of FPOM gathered from the outflowing water using continuous sediment samplers. The nutrient content of local leaf litter deposits, residence time of water in the springs and concentration of dissolved nutrients in spring sources and outflows were also measured to characterize the inputs and outputs of the studied system. The results show that headwater spring wetlands represent a significant source of high-quality FPOM for downstream river networks. The estimated concentration of FPOM (<1000 μm) in the 11 investigated springs was 3.1 ± 2.5 mg.L$^{-1}$. In general, the FPOM was relatively nutrient-rich (N = 19.25 ± 4.73 mg.L$^{-1}$; P = 2.04 ± 0.78 mg.L$^{-1}$; Ca = 9.65 ± 2.63 mg.L$^{-1}$; S = 4.07 ± 1.16 mg.L$^{-1}$; C = 278.68 ± 80.81 mg.L$^{-1}$). The C:N and C:P ratios in the local leaf litter deposits were higher than in FPOM (41.04 ± 14.32 vs. 14.70 ± 2.46 and 591.7 ± 168.83 vs. 154,77 ± 64,73, respectively), indicating that suspended FPOM is more nutritious for consumers. A significant trend in terms of size fractions of FPOM was identified: with decreasing C:N and C:P ratios particle size decreases as well. Overall, the data suggest that the relatively small helocrenes can serve as an organic matter transformers, receiving primary particles and dissolved organic matter, transforming them and favouring their transport downstream. These biotopes may represent a substantial discontinuity of the river continuum at its origin, important for nutrient dynamics and food supply of associated biotic communities.

cz/en/) KT, LP: IGA grant nr. 2010/42110/019,
Czech University of Life Sciences Prague (https://
www.czu.cz/en/). The funders had no role in study
design, data collection and analysis, decision to
publish, or preparation of the manuscript.

**Competing interests:** The authors have declared
that no competing interests exist.

## Introduction

The physical variables within a river system from the headwaters to the mouth are generally
changing along a continuous gradient. This gradient results in a continuum of biotic variables
and organic matter quality, quantity, utilization and transport along the length of a river [1].
Lotic systems are open and have a high capacity to retain nutrients from the catchment area
[2] especially smallest channels and headwaters can effectively transform the dissolved nutri-
ents into particulate matter [3]. Considerable work has been done to investigate organic matter
transport and nutrient cycling and processing in stream systems [2,4,5] and research into
organic matter cycling continues due to its theoretical and applied importance [6–10]. While
the continual downstream transport, i.e. nutrient spiraling and cycling, influences the abun-
dance and distribution of stream biota, the biota can, in turn, affect organic matter processing
and nutrient concentrations in the water [2].

One of the fundamental attributes of the continuous gradient of a stream ecosystem are
detrital pathways that determine the availability of organic matter as a food source for particu-
lar functional feeding groups in heterotrophic small streams systems [11]. According to River
Continuum Concept, mean detrital particle size decreases with increasing stream size [1],
although the supply of organic particles can be replenished from the riparian zone in down-
stream reaches [12] and the processes that transform organic matter may increase the size of
the particles by aggregation [13]. Headwater streams, especially those in forested landscapes,
are considered heterotrophic and depending primarily on coarse particulate organic matter
entering the stream from adjacent terrestrial communities [14]. Nevertheless, several studies
suggest that organic matter fluxes in stream ecosystems may be more complex than previously
supposed [15, 16]. For example, fine particulate matter (FPOM) export may be dramatically
affected by nutrient levels even in headwater streams [17, 18]. Additionally, extreme discharges
can substantially increase and invertebrate removal can substantially decrease the outflow of
FPOM [5].

Despite an increasing focus on organic matter fluxes in stream ecosystems, detrital path-
ways in spring areas remain relatively neglected. River Continuum Concept operates with the
idea of spring ecosystems with rapidly-emerging, nutrient-poor water, which is associated
with a limited biodiversity composed of species that can function on a restricted nutritional
base [1]. Such rheocrene-like springs (small streams of running water emerging directly at the
point of the source) [19] are, however, not very common in many areas. Instead, helocrene-
like springs (water seeping through a soil layer in a spring wetland) are often widespread in
moderately steep and non-forest landscapes [20, 21]. Helocrenes (spring wetlands, seepages)
[19] are, despite their abundance, often neglected in research because of their semiaquatic
character and the difficulty of location during the summer [22]. This type of spring ecosystem
often has considerable primary production of macrophytes and algae due to its semiaquatic
character [23]. Temperature fluctuations are limited and helocrenes typically do not freeze in
the winter. During the summer, their temperature increases only in open-water areas [24]. In
view of the spatial extent and specific conditions of helocrenes, this type of spring area may
have considerable effects on the downstream headwater systems. Nevertheless, the effect of
helocrenic spring areas on the nutrient cycles and the processing and transport of detrital par-
ticles has not been sufficiently investigated to date.

Organic matter in springs is generally plant detritus of various origins, size fractions, and
levels of decomposition [24]. Springs in forested watersheds are dominated by coarse particu-
late organic matter (CPOM) from leaf litter. In contrast, the leaf litter source in open-canopy
areas is compensated by instream primary production [25, 26]. The subsurface input of FPOM
can also be appreciable [27]. In addition to the particle size, which determines the possible

intake by consumers, their nutritional value is also critical [28]. Wood, leaf litter, and green leaves, like other terrestrial input, usually have substantially higher C: nutrient ratios than periphyton and FPOM [29]. Additionally, the C: nutrient ratios of organic matter are strongly influenced by the microbial colonization of the particles [30]. Generally, organic matter decomposition in benthic systems is accompanied by significant changes in the elemental composition of organic matter [30]. For filter-feeders, most of the energy income from this food source originates from the bacterial film of the detrital particle. Only a small fraction of the total energy income is from the particle itself, which is digestible only with difficulty [31]. This component is excreted and reused several times until it is completely decomposed [31]. Also, the nutrient value of FPOM is variously dependent on the site, season and ratio of the particles of different origin [32].

As a whole, these FPOM resources are considered to provide food that supports many functional groups of organisms in spring areas [22, 33–35], and most likely also supplies a substantial amount of organic matter to downstream ecosystems [26]. It has been documented that certain helocrenic springs may be a source of FPOM [27] or even that the concentration of FPOM in the spring source is higher than the concentration downstream [36]. Research on the possible role of spring-derived detritus from headwater streams downstream may thus help in understanding whole-system of organic matter dynamics, a critical topic for the conservation of freshwater ecosystems [11].

The aim of this study was to assess the role of helocrenic springs in terms of organic matter flows within a river network. We address this question by examining helocrene-like Central European springs. Specifically, we characterize the quality of FPOM (in three size fractions) exported from helocrenic springs, nutrient content, their concentration in the outflowing water and the microscopic physical components. The nutritional value of the constituent size fractions of detritus was evaluated based on the C: nutrient ratios together with simultaneously measured values of the dissolved nutrients, water retention times, and nutrient content in the leaf litter of the surrounding vegetation.

## Methods

### Study sites

Study area was located in the watershed of the Blanice River in the Šumava foothills, Czech Republic. The altitude of the area ranges from 790 to 1020 m above sea level; for a detailed description of the region, see. [37, 38]. The investigated springs have small temperature fluctuations (the differences between the winter and summer temperatures were 0.5 to 5.6 ˚C) and a neutral or slightly acidic pH (5.85 ± 0.16; mean ± SD). The springs were chosen from a set of undisturbed, permanent, and non-freezing springs listed in the database of springs maintained by the Agency for Nature Conservation and Landscape Protection of the Czech Republic (unpublished data). Because we aimed to include both nutrient-rich and nutrient-poor springs in our study, we selected the study sites by stratified random sampling on the basis of the nitrogen concentration in the outflowing water recorded by pilot sampling in 2007. The initial set of 92 springs with available information on $NO_{3-}$ concentration was scaled by the $NO_{3-}$ concentration gradient (N-rich springs: $NO_{3-} > 5$ mg.l$^{-1}$; N-poor springs: $NO_{3-} < 3$ mg.l$^{-1}$ and the studied set of springs was randomly selected from both the nitrate-rich (n = 6) and nitrate-poor (n = 5) sites. Concentration gradient in oligotrophic catchments is discussed in Černá et al. [39]. The springs were mostly in pastures, meadows or on the forest margin and rarely in the open-canopy forest (dominant trees in the surroundings were spruce, birch and willow). All springs were of the helocrenic type except two, which showed a transitional rheo-helocrenic character (running water directly from the point of the source, in combination with a

spring wetland). The slope of the sites was moderate to strong (9–30%). Selected springs have small temperature fluctuations (the differences between the winter and summer temperatures were 0.5 to 5.6 ˚C) and a neutral or slightly acidic pH (5.85 ± 0.16; mean ± SD). No permits and approvals were needed for the work in studied area.

## FPOM quality and quantity

We sampled the suspended FPOM draining from the 11 springs in four seasons (2009–2010): the spring sampling was performed in June after the growth of vegetation, summer sampling was conducted in August during the peak of vegetation growth, the autumn sampling in November after the decline of the vegetation and the leaf fall and winter sampling in February during a period of snow cover.

We used a modification of the approach of Cuffney and Wallace [40] adapted to our low-slope spring areas to capture FPOM with flow-through settlement vessels (Fig 1). The apparatus consists of a suction basket (polyethylene-PE-coated wire mesh– 10 mm loop size; inside the PE mesh– 1 mm loop size; Fig 1A), a flexible intake pipe (inside diameter 4 mm, polyvinyl chloride-PVC; Fig 1B), a settling vessel (volume 5 L, polyethylene terephthalate-PET; Fig 1C), equipped with an intake basket with openings limiting short-circuiting) and a discharge pipe (4 mm) with a PE mesh of 1 mm loop size in four layers, ending with a free tube (Fig 1D). The sampling apparatus is suited for small sloping water flows; it is based on connected vessels (gravity driven) and does not use external energy source. The detritus transported by the spring water was concentrated in the settling vessel during the exposure period. The size of the retained particles was restricted by a screen of 1-mm mesh mounted on the suction basket. We exposed these continuous sediment samplers in each spring area at the point where the outflow trickle below the helocrenic wetland was formed (6–43 meters from the spring's source point). The samplers were exposed for a period of 1 week during each season. We used the 1 week exposure period and two parallel samplers at each site because our pilot experiments showed that a long exposure increases the risk of clogging of the intake pipe with debris (clogged samplers were excluded from the analysis). If both devices functioned properly during the whole exposure period, we processed both samples and used the higher values for statistical analyses

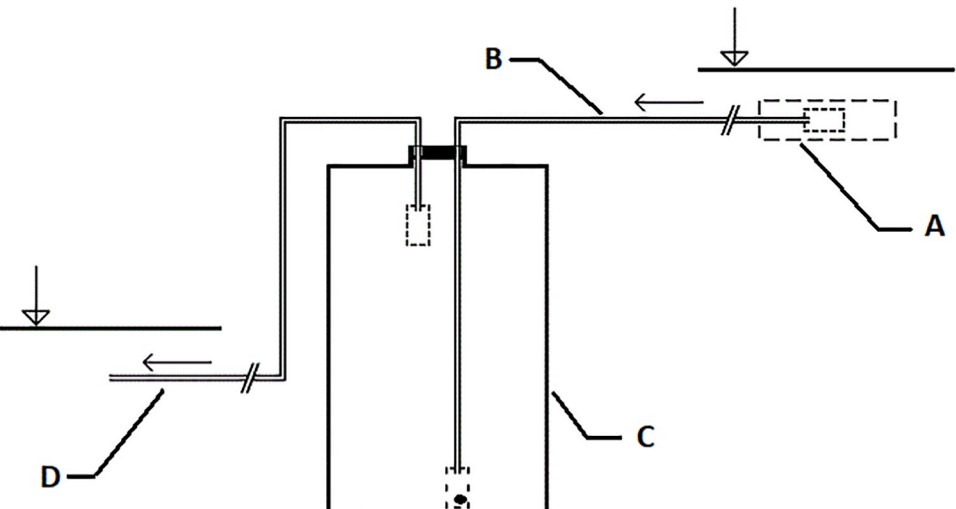

**Fig 1. Scheme of detrital sampler: suction basket (1), discharge pipe (2), settling vessel (3) discharge pipe (4).** The arrows indicate the difference of the water surface levels.

(measured as the total dry mass retained in each sampler). At the beginning and at the end of the sampling period, the average flow through the sampler was measured with a graduated cylinder and stopwatch. The samples were kept at 5–7 ˚C under dark conditions and processed within one week after the collection period.

In the laboratory, the detritus was pressureless sieved into three size fractions (1000–250 μm, 250–63 μm and smaller than 63 μm). Only the fine fraction was vacuum filtered. The fractions were dried at 105 ˚C, and weighed. The samples were analyzed, for C, N, P, S and Ca according to standard methods: determination of TOC in the solid matrix, Ca and P ICP-OES in the acidic leachate and N and S in a Variomax CNS analyzer [41, 42].

A microscopic analysis of the constituent size fractions was also done on samples collected in June 2010. The percentage of physical components (plant residues, fecal pellets and amorphous matter) was determined microscopically at 20 x– 100 x magnification according to Tichá et al. [24].

## Water discharge and FPOM concentration

A small spillway was installed under each spring outflow. The discharge was measured by trapping the water in a plastic bag for a defined time unit [21]. The discharge was measured twice during each period, namely, at the times of installation and removal of the continuous sampler. For further calculations, the average values were used to represent the sampling period because the fluctuations were small. The concentration and output of suspended detritus were estimated according to the following formulas:

detrital concentration $[\text{g.L}^{-1}]$

$$= \text{sum of dry mass of all three fractions of detritus } [\text{g}]/(\text{average flow through the sampler } [\text{L.hr}^{-1}] * \text{exposur } [\text{hr}])$$

$$\text{output } [\text{g.day}^{-1}] = \text{concentration } [\text{g.L}^{-1}] * \text{discharge of the spring } [\text{L.day}^{-1}]$$

## Retention time

The retention time of the studied helocrenes was measured by pouring a 1% solution of NaCl (5l, conductivity 1700 μS.cm$^{-1}$) into the spring source. The solution was poured slowly into the springs (approx. 1 min.) due to their low discharges. Simultaneously, the conductivity in the outflow was measured in a continuous manner. The peak value of the conductivity was used as a measure of the retention time [43]. The values were measured during the spring (April, before the vegetation growth) and summer (August, when vegetation cover was well developed).

## Leaf litter quality

The leaf litter from the spring surface was sampled in November 2009. Two squares, each with an area of 1 m$^2$, were marked on each spring surface, and all the leaf litter was removed. In the laboratory the samples were air-dried, separated to genera, and separately weighed as the total dry mass of each tree species. Then they were merged into the original samples. For comparison with the dry mass of detritus, we used data of leaf litter dry mass. One-tenth of dry mass of every sample was then pulverized and chemically analyzed using techniques identical to those used with the detritus samples.

## Water chemical analysis

Samples of water from the source and outflow of the springs were taken simultaneously before the installation of the FPOM samplers. $NO_3$, ortho-$PO_4$ and Ca were determined according to standard methods (ISO 11885:2007) in the laboratory.

## Statistical analyses

Paired *t*-tests were used to test the differences in water quality between the source and outflow sites of the studied set of springs (data from all sampling occasions were pooled). The nutrient content of each size fraction of FPOM (dependent variables) was analyzed with a general linear model (GLM) with season and site (explanatory variables). If necessary, the data were log-transformed to meet normality assumptions. All statistical analyses were performed with R 3.5.2 software (R Development Core Team 2018).

# Results

## Detritus quality and quantity

FPOM samplers captured, on average, 2610 mg of dry mass (min = 68, max = 12,864). In 88 1-week measurements, the sampler was clogged only eight times. The samples from the clogged samplers were excluded from the analyses.

The average discharge of 11 selected springs was $0.53 \pm 0.37$ L.s$^{-1}$ (mean $\pm$ SD). The concentration of FPOM was $3.10 \pm 2.5$ mg.L$^{-1}$. The specific concentrations for the separate fractions were $0.85 \pm 1.75$ mg.L$^{-1}$ (1000–250 µm, coarse fraction), $1.93 \pm 1.36$ mg.L$^{-1}$ (250–63 µm, intermediate fraction) and $0.32 \pm 0.28$ mg.L$^{-1}$ ($< 63$ µm, fine fraction).

The microscopic analysis of physical components identified three principal constituents: plant residues, faecal pellets of macroinvertebrates and amorphous matter. The amorphous matter was most likely composed of very fine coagulated detritus with bound microorganisms. On average, the coarse fraction (N = 18) consisted of $17.4 \pm 6.7\%$ plant residues, $20.7 \pm 13.9\%$ faecal pellets and $61.9 \pm 15.4\%$ amorphous matter of estimated volume. The intermediate fraction (N = 16) consisted of $42.5 \pm 20.8\%$ plant residues, $37.5 \pm 21.8\%$ faecal pellets and $20.0 \pm 12.6\%$ amorphous matter of estimated volume. A microscopic analysis of the fine fraction was not performed due to the use of vacuum filtration.

The GLM model incorporated the effects of site, sampling season and size fraction. The nutrient contents of the three size fractions of FPOM were significantly different for all measured elements, except of P (Fig 2, Table 1). There were also significant differences in detritus nutrient content between sampling seasons for C ($p<0.001$) and for P, Ca, and dry mass ($p<0.01$) (Table 1) with highest mean nutrient content recorded in November. For N and S the season was not significant. All measured elements concentration significantly differed among sites (all $p<0.001$). The comparison of various size fractions, showed significant differences of nutrient ratios for C:P, C:Ca, and N:P ($p<0.001$) and for C:N ($p<0.01$). For C:S the results were not significant (Fig 3, Table 1). In general, the content of C, N, Ca and S decreased with decreasing particle size. However, such a dependence was not observed for phosphorus. The C: nutrient (except C:S) ratios also decreased with the particle size.

## Water chemical analysis and retention time

In general, the mean values of dissolved nutrients ($NO_3$, o-$PO_4$) were higher just in the source, compared to the helocrenic spring outflow. While in the source the $NO_3$ and o-$PO_4$ concentrations were $5.8 \pm 5.5$ mg.L$^{-1}$, resp. $0.06 \pm 0.03$ mg.L$^{-1}$, in the outflow it was $4.7 \pm 4.7$ mg.L$^{-1}$,

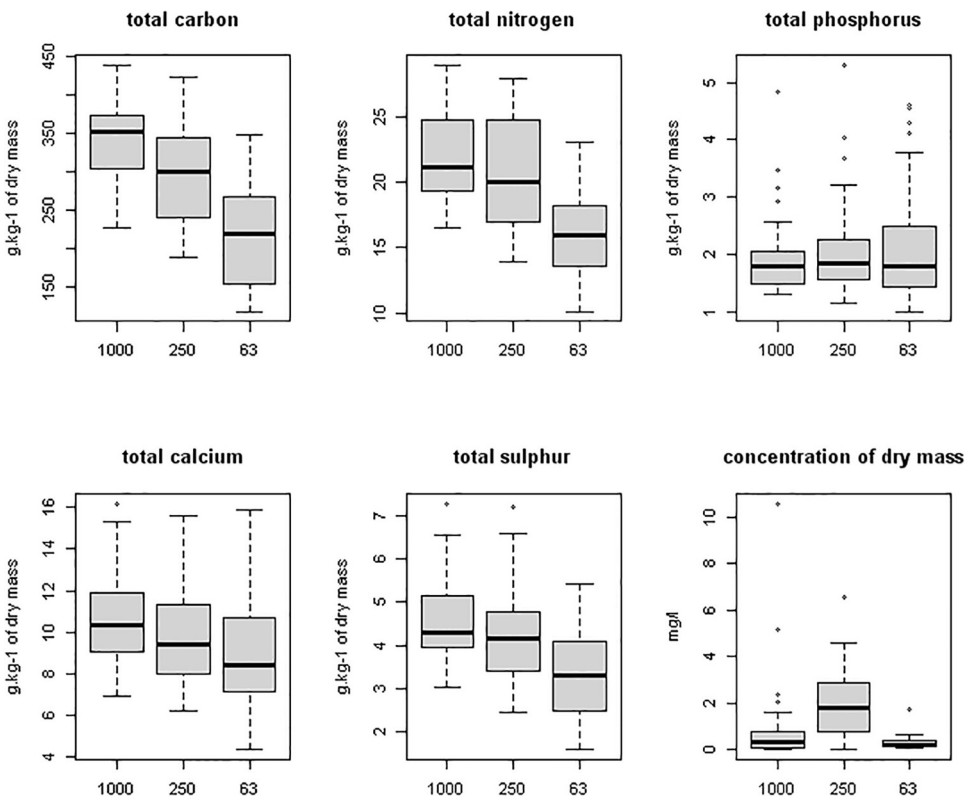

**Fig 2. Nutrient content and the estimated dry mass concentration of the constituent size fractions of detritus sampled from outflowing spring water.** Median, 1st and 3th quartile, extremes and outliers are indicated.

resp. $0.04 \pm 0.03$ mg.L$^{-1}$. No difference between the source and the outflow was found for Ca (Table 2).

The retention time was successfully measured in eight springs (the concentration of tracer in the outflow water was undetectable in three springs). The retention times varied markedly.

**Table 1. The effects detritus fraction size (1000–250 μm, 250–63 μm, < 63 μm) and sampling season (spring: Before the vegetation grow, summer: Maximum of vegetation grow, autumn: After leaf falling, winter: Snow cover) on nutrient content (C, N, P, Ca and S), detritus concentration, and C:N, C:P, C:Ca and C:S ratios determined by GLM in detritus samples taken from outflowing spring water.**

| Parameter | Fraction (DF = 2) | | Season (DF = 3) | |
|---|---|---|---|---|
| | F | P | F | p |
| N | 54.3 | <0.001 | 0.4 | n.s. |
| P | 0.7 | n.s. | 4.2 | <0.01 |
| S | 34.4 | <0.001 | 0.7 | n.s. |
| Ca | 17.8 | <0.001 | 5.5 | <0.01 |
| C | 102.8 | <0.001 | 10.6 | <0.001 |
| Sus. | 55.3 | <0.001 | 4 | <0.01 |
| C/N | 5.7 | <0.01 | 6.2 | <0.001 |
| C/P | 50.1 | <0.001 | 1.2 | n.s. |
| C/S | 2.2 | n.s. | 9.1 | <0.001 |
| C/Ca | 49.9 | <0.001 | 5.1 | <0.01 |
| N/P | 38.7 | <0.001 | 2 | n.s. |

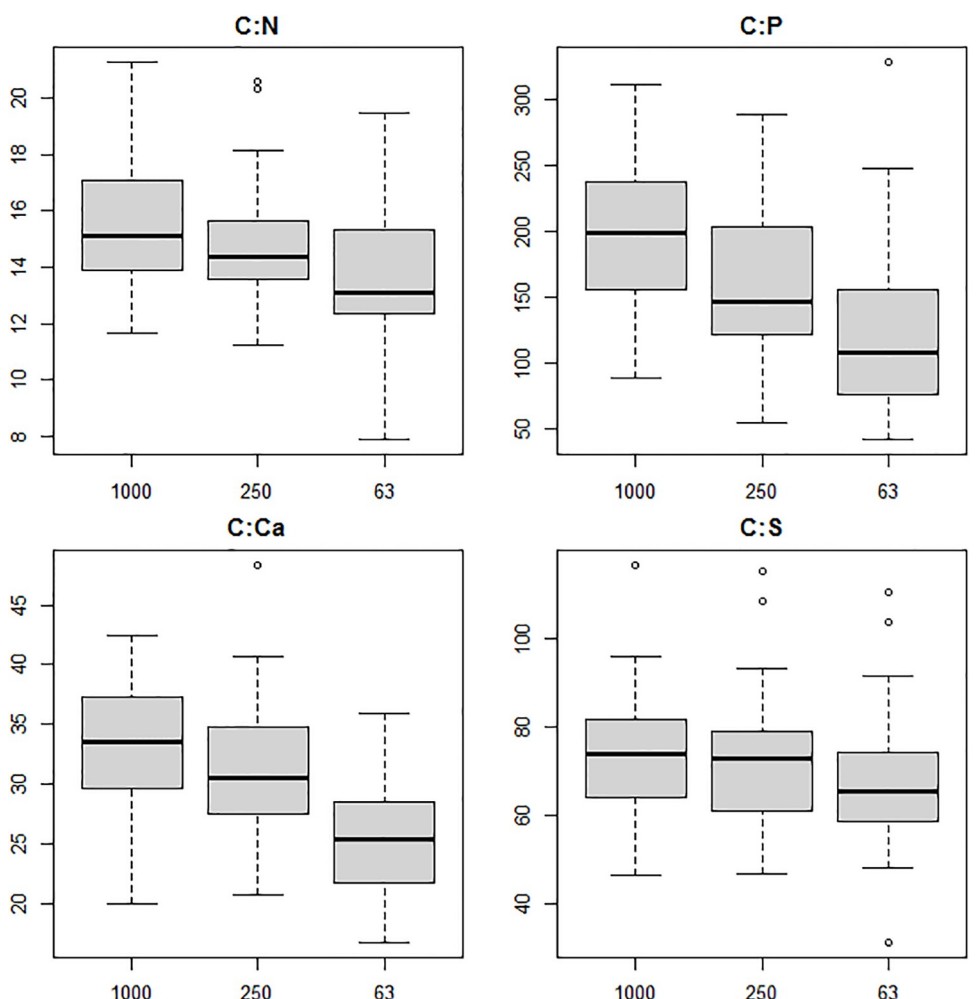

**Fig 3. C:N, C:P, C:Ca and C:S ratios in the constituent size fractions of detritus.** Median, 1st and 3rd quartile, extremes and outliers are indicated.

**Table 2. Dissolved nutrients in the spring source and outflow and the significance of paired t-tests of these data (N = 44).**

| Parameter | Position | Mean ± SD | Range | t | DF | p |
|---|---|---|---|---|---|---|
| | | (mg/l) | (mg/l) | | | |
| $NO_3$ | source | 5.8 ± 5.5 | 0.7–19.1 | 2.8 | 43 | <0.01 |
| | outflow | 4.7 ± 4.7 | 0.1–16.4 | | | |
| o-$PO_4$ | source | 0.06 ± 0.03 | 0.01–0.13 | 6.38 | 43 | <0.001 |
| | outflow | 0.04 ± 0.03 | 0.0025*–0.11 | | | |
| Ca | source | 6.1 ± 3.0 | 3.2–15.7 | -0.03 | 43 | n.s. |
| | outflow | 6.2 ± 3.4 | 2.4–19.4 | | | |

* under detection limit

It was very short (3–14 min) in stronger springs with a clear visible outflow trickle and a small wetland component and noticeably longer (33–41 min) in weak springs with an extensive wetland component. The average velocity of the water passing through the spring was $0.024 \pm 0.021$ m.s$^{-1}$.

## Leaf-litter quality

The amount of leaf litter deposited in the springs was $43.0 \pm 44.7$ g.m$^{-2}$ (mean and SD). Overall, the leaf litter was dominated by deciduous trees: birch (23.1%), willow (18.7%), beech (15.6%) and maple (9.5%), accompanied by less frequent deciduous trees such as alder, cherry tree, aspen, rowan and lime (4.8%, 4.7%, 3.6%, 1.6% and 0.1%, respectively). Coniferous pine (7.5%) and spruce (3.6%) litter, as well as grasses (8.3%) and other herbs (2.6%), were also present.

The average content of nutrients in the leaf litter at the spring surface (Table 3) did not show so much variability as nutrient content in detritus. The content of basic nutrients was for C = 541 ± 30, for N 14.87 ± 5.61 and for P = 0.968 ± 0.223 g/kg. The calculated values of the nutrient ratios in the leaf litter were C:N 37.1 ± 14.0, C:P 548.4 ± 167.7, C:Ca 39.8 ± 11.7 and C:S 403.9 ± 119.4.

## Discussion

### Sources of FPOM

Since the introduction of the River Continuum Concept [1], many studies of the properties and transport of detrital particles have been conducted in first- and second-order headwater streams. Surprisingly, a very small number of studies have focused on FPOM in spring areas. Suspended FPOM is sometimes considered to be absent from springs. According to this view, the headwaters below the springs are essentially supplied by allochthonous matter. Headwater streams are predominantly viewed as accumulators, processors and transporters of materials from terrestrial systems [1, 44, 45]. In contrast, this study indicates that a substantial amount of FPOM can originate directly from wetland springs.

The average concentration of suspended FPOM in the water was estimated to be 3.1 mg.L$^{-1}$ in our samples, whereas the average concentration of FPOM in a typical second-order stream in a forested catchment was calculated as 2 mg.L$^{-1}$ [46]. The principle that the particle size of the transported organic material should become progressively smaller along the continuum [1] implies that the concentration of the fine particles will be lowest below the spring. However, substantial concentrations of all size fractions of FPOM were recorded below all the wetland springs that we investigated. The most well-represented fraction was the intermediate fraction (250–63 μm), with a concentration of 1.93 mg.L$^{-1}$. Moreover, because we used a sediment sampler system, only the depositable particles were successfully recorded in our experiments, and our estimates of the total amount of FPOM should be considered conservative. We

**Table 3. Average content of nutrients in leaf litter deposits at the spring surface (N = 22).**

| Parameter | Mean ± SD | Range |
|---|---|---|
| | (g/kg) | (g/kg) |
| C | 541 ± 30 | 488–616 |
| N | 14.87 ± 5.61 | 8.5–30.3 |
| P | 0.968 ± 0.223 | 0.552–1.4 |
| Ca | 14.1 ± 3.6 | 8.77–21.1 |
| S | 1.32 ± 0.32 | 0.89–1.97 |

suggest that, due to the low deposition rates and long transport distances of small particles of FPOM in mountain streams [47], the particles from springs may travel relatively long distances and are used as a food source for downstream benthic communities.

Large volumes of leaf litter input are produced in the riparian zones of forested headwaters. This production fuels the spiraling of organic matter down the river continuum [10]. Hence, leaf decomposition contributes large amounts of FPOM to the stream's detrital supply [48, 49] although other sources of FPOM could be of equal or greater magnitude in some systems [50]. These sources are probably also very important in case of helocrenic springs we examined. FPOM can arise from dissolved organic matter (DOM) by physicochemical processes, such as flocculation and adsorption [13], or by microbial uptake [51]. There is little information about the sloughing of algal mats and other organic layers or about the role of detritus produced from the floor litter and the soil [50, 52]. Obviously, many of these sources could form detrital particles just before the water emerges from the ground. In the work of Iversen [27], the groundwater input of FPOM in one spring was determined as 68 kg.yr$^{-1}$. Our observation show that the fine detritus in helocrenic springs can also in part emerge just from the groundwater, because in some sampled springs, fine detritus was collected directly at the spring source.

In addition to the groundwater processes, it is necessary to consider the processes taking place in the springs. In general, the transport rates of detrital particles in streams (measured for three categories: sticks, leaves and FPOM) are substantially higher than the breakdown rates for these types of particles [46]. For this reason, an FPOM particle on the stream bottom is more likely to be transported downstream than to be decomposed [46]. Our results show, that the transport velocity of FPOM in wetland springs can be lower than in the average headwater stream, as referred in Newbold et al. [2]. The spring area, vegetation cover, slope and other variables vary in ways that produce water retention periods of considerably different lengths. In the studied helocrenes, the estimated retention period varied from 3 to 41 minutes. Despite this variation, the retention times in spring wetlands are markedly longer then in the reaches downstream. As a result, the relatively small helocrenes could serve as "detrital reactors", receiving inorganic nutrients, DOM and primary particles from the subsurface, transforming them and transporting them as FPOM downstream. These processes represent a typical example of highly effective transformations of both dissolved and particulate nutrients in the primary river network at basic runoff as described by the River Network Saturation concept [45]. Naturally, the leaf litter and woody debris also enter these "detrital reactors". However, we suppose that the input of these allochthonous material into our springs is variable and relatively low because the study sites were primarily on pastures or forest margins and only a few of them were in the forest (we recorded only 43.0 ± 44.7 g.m$^{-2}$ of leaf litter deposits for the autumn study season).

In the forested or partially forested catchments the input of organic matter shows a strong seasonal variability [27, 32]. But, the amount of deposited organic matter in the stream is not very different in the non-forested catchments compared to the catchment covered with the deciduous forest [53]. Non-forested catchments with higher primary production, dwelled by rich planktonic and benthic community can produce more faecal pellets and parts of macrophytes [16]. High production of FPOM in our springs with a low input of allochthonous organic matter, is likely subsidized with organic material from the subsurface and primary production.

## Nutritional value of FPOM

In general, it is considered that the nutritional value of FPOM in headwater streams is low, and consumers must consume a large amount to meet their nutritional needs [48, 54]. The

variability of nutrient content in FPOM particles is remarkable. For example, a FPOM particle from the stream bottom is qualitatively different from a fresh FPOM particle from the water column [14]. The FPOM particles from second-order streams with forested watersheds are created from refractory material [46]. This material is not as easily decomposed as the leaf litter. These refractory characteristics enable the relatively long-distance transport of these particles to higher-order streams [46]. The content of dissolved nutrients in this type of headwater stream is correspondingly low [17]. In contrast, our study area, with a predominantly treeless sloping landscape, represents a different situation. The helocrenic-spring water right in point of source contains markedly more dissolved nutrients than the water in the downstream brooks and rivers (Table 2). Dissolved nutrients in downstream rivers of the same area were studied by Černá et al. [39]. In contrast to the results of previous studies [17, 30], FPOM immediately below the helocrenic springs is relatively rich in nutrients.

Our study found a significant difference in nutrient content among the three constituent size fractions measured; the finest fraction had the lowest concentration of nutrients, except of phosphorus that prevails mostly in anionic form and is chemically bound in different way. We suppose that with finest fraction the inorganic substances predominate. The size-fractions used in our study and the measured flow velocity enable the entraining and transport of silt and finer particles as shown by the Hjulström curve [55]. On the other hand, the C: nutrient ratios decreased significantly with decreasing particle size, a result that is in good agreement with previous studies [56]. The smaller particles, with their lower C: nutrient ratio, could be more favorable for consumers, whereas the larger fractions serve as a "storage" compartment for food. Most likely, this stored food could be used by filter feeders in downstream reaches after this material is further processed. Decreasing particle size is accompanied by a relative increase in surface area and thus by a higher capacity for microbial colonization and nutrient adsorption [30].

The comparison of C:N and C:P ratios in the recorded autumn leaf litter deposits and in suspended FPOM below the springs shows lower ratios (higher nutritional value) of both the characteristics in FPOM. Obviously, detritus exported from the helocrenic springs originates not only from poorly decomposable remains but also from other components, such as the nutrient-rich microbial film [18]. The average leaf C:N found by our study is in good accordance with previous studies [30, 57], but the detrital C:N is substantially different. In Sollins et al. [57], heavy detrital particles with a low C:N ratio included a substantial amount of crystalline mineral matter and had a C:N similar to floodplain soil. It is probable that the particles in our samples also originate, in part, from soil horizons, but they are most likely mixed with other particles with a higher C content because the C:N was almost twice as high, compared to floodplain soil. Conversely, in Cross et al. [30] the C:N and C:P values for all size fractions examined were generally higher than those found by our study. In Cross et el. [30], the C: nutrient data were based on an extensive search of literature sources, and the analyzed FPOM originated from different freshwater systems. Therefore, we consider that the lower C:N found by our study was specific for helocrenes. However, it appears evident that substantial variability occurs among catchments and is, most likely, caused by differences in local conditions. In this case, we consider that the helocrenic spring type and its specific environmental conditions and ongoing chemical processes is the critical factor.

## Concepts and conclusions

In general, we found large amounts of fine, nutrient-rich detritus in the helocrenic springs examined. These findings are further supported by the presence of invertebrate assemblages [22] typical of up to third- to fourth-order streams, a result that contrasts with the assumptions

of the River Continuum concept (RCC) [1]. This widely used concept describes only processes in some types of springs, particularly rheocrenes [58]. Later concepts correspond better with detrital processes in helocrenic springs. This type of small but ecologically very important ecotone environment needs specific theoretical framework [59]. In general, helocrenes are not directly connected to the river network and they are often only remote member of complex riverine-hyporheic-groundwater system [59, 60], However, their importance for local biodiversity is essential [58].

The stream hydraulics concept (SHC) [61] represents one possible explanation of detrital processes in helocrenes. The SHC invokes water velocity as the main controlling factor of organic matter distribution. Slowly flowing lowland rivers allow the accumulation of fine sediments and the occurrence of sediment burrowers. Concurrently, the low discharge rates in helocrenic springs and long residence time of water allow the accumulation of FPOM and occurrence of specific invertebrate assemblages.

Helocrenes represent a semi-aquatic environment that is strongly bound to groundwater. Therefore, not only longitudinal connectivity but also the connection with the hyporheal environment (both the anoxic organic bed and shallow subsurface oxygenated groundwater) and the transverse linkage to the banks and their wetland and terrestrial assemblages becomes important, as previously described in the "hyporheic corridor concept" [12]. The influence of this vertical and lateral connectivity in helocrenes with low velocity can outweigh the longitudinal connectivity from the source to the outflow trickle. It appears that as well as in large rivers [62], processes and nutrient cycles on the microbial level (autotrophic bacteria, heterotrophic bacteria, bacteriophages, protozoans) also play a crucial role at the very beginning of the river network, i.e., in the metabolism of specific environment of springs. This principle is also supported by the marked decreases in dissolved forms of nitrogen and phosphorus reported in our study. Similar results from large New Hampshire headwater wetlands have been published by Flint and McDowell [63]. General view of the connection of spring wetlands and downstream waters provides a comprehensive review study by Alexander et al. [64], describing retention and transformation of nutrients as one of the crucial functions of nonfloodplain wetlands.

As described in our study, input of nutrient-rich fine particles into first-order streams can be frequent in diverse biotope types. In a global context, however, a great portion of the primary river network features open canopies, either for natural reasons (tundra or semiarid regions) or due to the increasing pressure of human populations (pasture grasslands, meadows) [53]. Therefore, a typical origin of a river net might not have forest rheocrenic characteristics, but could instead be a spring wetland, e.g. helocrene. Because of their considerable semiaquatic character, helocrenes are often overlooked in hydrobiological and ecological research. However, they are very common in open landscapes with gentle slopes [22]. In the area of about 60 km$^2$ where we were doing the research about 1 100 springs has been mapped [65]. If only half of them were helocrenes connected to the river network, their theoretical production of FPOM would be about 28 498 kg of dry mass per year. What is the real production of these springs and how big is their impact on spring and downstream ecosystems is still unknown. But it is necessary to take the spring production into account in the annual organic matter budgets. Consequently, from a global perspective, these specific biotopes may represent a relatively common discontinuity of the river continuum at its origin. This characteristic would have important implications for the functions of organic matter pathways within the river network, as recently noted by several authors [11, 66, 67]. Research on the processes of downstream transport and utilization of nutrients still evoke new questions. We wanted to explain why a relatively large amount of nutrients in the form of FPOM appears already in the headwaters. If in the watersheds of these flows were many spring

wetlands, it is likely that FPOM originates from these springs, rather than from the decomposition of leaf litter.

## Acknowledgments

We thank Vojtěch Mrázek, Věra Kladivová, Josef Rebec, Michal Bílý, Josef Fuksa, Michal Horsák and Jaroslav Hruška for inspiration, help in the field and comments on the earlier versions of the manuscript. The Agency for Nature Conservation and Landscape Protection of the Czech Republic provided us a source database of springs collected by Bohumil Dort. Also, we thank two Plos One reviewers: Marco Cantonati and the second anonymous reviewer.

## Author Contributions

**Conceptualization:** Ondřej P. Simon, Karel Douda.

**Data curation:** Ondřej P. Simon.

**Formal analysis:** Kamila Tichá, Jakub Houška.

**Funding acquisition:** Ondřej P. Simon, Lucie Peláková.

**Investigation:** Kamila Tichá, Lucie Peláková.

**Methodology:** Ondřej P. Simon, Jakub Houška, Karel Douda.

**Project administration:** Lucie Peláková.

**Resources:** Kamila Tichá, Jakub Houška.

**Software:** Karel Douda.

**Validation:** Kamila Tichá, Ondřej P. Simon, Lucie Peláková.

**Visualization:** Kamila Tichá, Karel Douda.

**Writing – original draft:** Kamila Tichá, Ondřej P. Simon.

**Writing – review & editing:** Kamila Tichá, Ondřej P. Simon, Karel Douda.

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
