## [Decision Letter · Decision Letter 0]

8 Jan 2020

PONE-D-19-31487

The reversal of a river continuum at the origin: nutrient-rich fine particulate organic matter exported from helocrene springs

PLOS ONE

Dear Dr. Ticha, 

Thank you for submitting your manuscript to PLOS ONE. After careful consideration, we feel that it has merit but does not fully meet PLOS ONE’s publication criteria as it currently stands. Therefore, we invite you to submit a revised version of the manuscript that addresses the points raised during the review process.

We would appreciate receiving your revised manuscript by Feb 22 2020 11:59PM. To enhance the reproducibility of your results, we recommend that if applicable you deposit your laboratory protocols in protocols.io, where a protocol can be assigned its own identifier (DOI) such that it can be cited independently in the future. For instructions see: http://journals.plos.org/plosone/s/submission-guidelines#loc-laboratory-protocols

We look forward to receiving your revised manuscript.

Kind regards,

Kai Yue, Ph.D.

Academic Editor

PLOS ONE

Additional Editor Comments:

I have now received the comments from two reviewers, both of them think the manuscript is of high quality, but one reviewer suggested a major review before the MS can be accepted for publication.

Journal Requirements:

3. Please upload a copy of Figure 4, to which you refer in your text on page 7. Please also include a caption for figure 4.  If the figure is no longer to be included as part of the submission please remove all reference to it within the text.

Reviewers' comments:

Reviewer's Responses to Questions

**Comments to the Author**

1. Is the manuscript technically sound, and do the data support the conclusions?

Reviewer #1: Yes

Reviewer #2: Yes

2. Has the statistical analysis been performed appropriately and rigorously? 

Reviewer #1: Yes

Reviewer #2: Yes

3. Have the authors made all data underlying the findings in their manuscript fully available?

Reviewer #1: Yes

Reviewer #2: No

4. Is the manuscript presented in an intelligible fashion and written in standard English?

Reviewer #1: Yes

Reviewer #2: Yes

5. Review Comments to the Author

Reviewer #1: This paper provides useful novel data on FPOM in a specific spring type (helocrenic springs = seepages). My main observation is that the Authors insist in presenting their results in relation to the River Continuum Concept (RCC) whilst they largely ignore the abundant literature showing that springs are special freshwater habitats with unique characteristics. I also have other few major and several minor observations which are listed below, and I am glad to recommend the paper for publication in PLOS ONE once these revisions have been carried out.

Major observations:

- The RCC is not the most suitable theoretical framework to understand FPOM in springs because it views running water systems as necessarily connected and fed at their origin by a specific, though widespread, spring type, i.e. flowing springs (= rheocrenic springs). Springs are special freshwater systems that can belong to a variety of types, and they can often be disconnected from the running water system, i.e. isolated.

- On the contrary, the Authors largely ignore theoretical frameworks specifically developed for springs, such as marked heterogeneity of springs (e.g., Freshwater Sci. 31, 463–480, 2012), spring types (e.g., Hydrogeol. J. 17 (1), 83–93, 2009; Journal of Limnology 70(1s), 147-154, 2011; Ecological Indicators 110. DOI: 10.1016/j.ecolind.2019.105803, 2020), springs as multiple ecotones (e.g., Freshwater Sci. 31, 463–480, 2012).

- Helocrenic springs are a specific kind of springs and are indeed defined as springs that resemble wetlands. Though interesting and useful, it is thus not too much surprising to find that seepages produce more FPOM than subsequent small mountain streams.

- Lines 438-444: Yes: springs are multiple ecotones!

- Title: I would detach the discussion of the results from the RCC, and would thus also propose to change the title. Moreover, Authors should be careful in using the term “reversal” which actually doesn’t seem to apply to their findings, and that was recently used in publications on springs in Florida to indicate an effective change in the direction of flow (these springs were close to and feeding rivers; due to water overdraft, spring discharge decreased to such an extent that river water now enters into the springs causing nutrient enrichment).

- Proposal for a new title: “Seepages (helocrenic springs) as sources of fine particulate organic matter in small foothill drainage basins”.

- Temperatures should always be reported with a space between the value and the Celsius degree unit.

Minor observations:

- Terminology: Please note that ‘helocrenes’, ‘rheocrenes’ etc. should be used as nouns whilst ‘helocrenic’, ‘rheocrenic’ etc. should be used as adjectives.

- Line 52: “continuous gradient”: not always… there are many natural discontinuities, and springs in particular can be very isolated.

- The language is already good but needs some editing, in particular in important parts of the text such as the Abstract and the Discussion. In an attempt to help to accomplish this, I provide the following edits:

• Lines 42-44: Change to “A significant trend in terms of size fractions of FPOM was identified: with decreasing C:N and C:P ratios, particle size decreased as well.

• Line 46: Change to “them and favouring their transport downstream”.

• Line 59: Change to “i.e. nutrient spiraling and cycling influence the…”.

• Lines 64-65: Change to “According to the River…”.

• Line 77: Change to “neglected. The River…”.

• Line 78: Insert comma after “rapidly-emerging”.

• Line 85: Change to “the difficulty of location”.

• Line 87: Change to “and helocrenes typically do not freeze”.

• Line 109: Change to “these FPOM resources are”.

• Line 110: Insert comma after “32-34]”.

• Line 165: Change to “gravity driven”.

• Line 172: Change to “increases”.

• Line 176: Change to “and at the end”.

• Line 178: Change to “one week”.

• Line 210: Change to “before vegetation growth) and summer (August, when vegetation cover was…”.

• Line 228: “t-tests”: the “t” should be in Italics (see also Tab. 2).

• Line 279: Change to “in the source”.

• Line 294: Change to “Leaf-litter quality”.

• Tab 3: Period (dot) at the end of the legend is missing.

• Line 319: Change to “originate directly”.

• Line 344: Change to “can also in part emerge just”.

• Line 351: Change to “than to be decomposed”.

• Line 356: Change to “longer than in the reaches”.

• Lines 376-377: Change to “FPOM in headwater streams is low”.

• Line 386: Change to “The helocrenic-spring water”.

• Line 391: Change to “assemblages in helocrenes… do not correspond”.

• Line 393: Change to “P. personatum”.

• Line 394: Change to “we consider the helocrenic spring”.

• Line 398: Change to “bound in a different way”.

• Line 400: Change to “measured flow velocity enable”.

• Line 401: Change to “as shown by the”.

• Line 402: Change to “with decreasing particle size,”.

• Line 423: Change to “for helocrenic spring areas”.

• Line 425: Change to “consider the helocrenic spring type”.

• Lines 436-437: Awkward sentence: Please re-write.

• Lines 453-454: Unclear sentence. Do you mean: “Most likely, input of nutrient-rich fine particles into first-order streams can be frequent in diverse biotope types.”

• Line 454: Avoid repetition: “primarily”… “primary”.

• Line 456: Change to “features”.

• Line 459: Change to “could instead be a spring wetland, i.e. a helocrene”.

• Line 462: Change to “research about 1100 springs have been mapped”.

• Line 465: Change to “ecosystems is still”.

• Line 473: Change to “flows there are many”.

• Line 568: “Pisidium” should be in Italics.

Reviewer #2: The paper describes the influence of fine particulate organic matter (fpom) from spring regions within fluvial network systems respectively on downstream headwater reaches. This research topic is basically needed, because it is very poor understood out- and inputs between spring regions and the downstream headwater region. The results show that headwater springs represent a significant source of high-quality FPOM for downstream river networks. 11 studied springs might not seems very representative to conclude overall processes, but the results gave a first insight of the role of springs within stream networks and its nutrients budget. small helocrenes can serve as an organic matter transformers, receiving primary particles and dissolved organic matter, transforming them and transporting them downstream. That shows how important it is to study spring region within the river continuum concept and within the ecology of fluvial network systems. The study area is located in the watershed of the Blanice River in the Šumava foothills of the Czech Republic. Helocrene springs are representative for Low Mountain Ranges in Central Europe. Unfortunately, the springs were not investigated for a period longer than one year or 4 seasons within one year. Longer time series would be desirable in order to investigate dynamics and fluctuations that can be traced back to flow regimes in particular. However, the sampling and laboratory methods are also very time-consuming and sufficient for a first overview. A very interesting result of this study indicates that a substantial amount of FPOM can originates directly from wetland springs. This is of high importance, because springs or headwater regions are predominantly viewed as accumulators, processors and transporters of organic materials from terrestrial systems. Ecologically interesting is that the particles from springs may travel relatively long distances and are used as a food source for downstream benthic communities. It is very important and good that forest and non-forest springs have been investigated and that we get here a comparison regarding the FPOM budget. Furthermore, it is interesting that from the field observation data in this study (and not from different literature based data) a lower C:N ratio for springs was found. Studying spring regions is of importance for nutrient ecology. The conclusion of the discussion is evident: It is to consider that the helocrene spring type and its specific environmental conditions and ongoing chemical processes as the critical factor. Thank you for that brilliant paper.

6. PLOS authors have the option to publish the peer review history of their article (what does this mean?). If published, this will include your full peer review and any attached files.

Reviewer #1: Yes: Marco Cantonati

Reviewer #2: No

---

## [Author Response · Author response to Decision Letter 0]

6 Mar 2020

Dear Editor:

We send a revised paper with previous title "The reversal of a river continuum at the origin: nutrient-rich fine particulate organic matter exported from helocrene springs", which was renamed to “Helocrenic springs as sources of nutrient rich fine particulate organic matter in small foothill watersheds”.

We included the suggestions of reviewers and editor’s comments (please see below) and we hope that the manuscript is prepared for publication. Please do not hesitate to contact us if you had any questions or requests.

Best regards,

Kamila Tichá

corresponding author

kamila.ticha@vuv.cz, +420 220 197 367

Editor Comments:

Reply: Yes, the style requirement were incorporated to the revised manuscript

• Please upload a copy of Figure 4, to which you refer in your text on page 7. Please also include a caption for figure 4. If the figure is no longer to be included as part of the submission please remove all reference to it within the text.

Reply: On page 7 we describe the sampling device, pictured in Figure 1. The designation Fig.1-1 to Fig. 1-4 described individual parts of this apparatus; the parts were numbered 1, 2, 3, 4 in the picture. We recognized that this numbers were confusing for the reader, so now we prefer to use letters: A, B, C, D. In the text it was changed to Fig. 1-A, Fig. 1-B etc.

Reviewer #1 Comments:

Major observation:

• The RCC is not the most suitable theoretical framework to understand FPOM in springs because it views running water systems as necessarily connected and fed at their origin by a specific, though widespread, spring type, i.e. flowing springs (= rheocrenic springs). Springs are special freshwater systems that can belong to a variety of types, and they can often be disconnected from the running water system, i.e. isolated.

• On the contrary, the Authors largely ignore theoretical frameworks specifically developed for springs, such as marked heterogeneity of springs (e.g., Freshwater Sci. 31, 463–480, 2012), spring types (e.g., Hydrogeol. J. 17 (1), 83–93, 2009; Journal of Limnology 70(1s), 147-154, 2011; Ecological Indicators 110. DOI: 10.1016/j.ecolind.2019.105803, 2020), springs as multiple ecotones (e.g., Freshwater Sci. 31, 463–480, 2012).

Reply:

We understand the reviewer's request and have decided to change the title and a part of the discussion. We did not intend to ignore any previous studies nor the theoretical framework regarding the springs (please see details below).

• Title: I would detach the discussion of the results from the RCC, and would thus also propose to change the title. Moreover, authors should be careful in using the term “reversal” which actually doesn’t seem to apply to their findings, and that was recently used in publications on springs in Florida to indicate an effective change in the direction of flow (these springs were close to and feeding rivers; due to water overdraft, spring discharge decreased to such an extent that river water now enters into the springs causing nutrient enrichment).

• Proposal for a new title: “Seepages (helocrenic springs) as sources of fine particulate organic matter in small foothill drainage basins”.

Reply:

Yes, we discussed the title change and decided to do so. The new title is “Helocrenic springs as sources of nutrient rich fine particulate organic matter in small foothill watersheds”.

Temperatures should always be reported with a space between the value and the Celsius degree unit.

Reply: Done

Minor observations:

• Terminology: Please note that ‘helocrenes’, ‘rheocrenes’ etc. should be used as nouns whilst ‘helocrenic’, ‘rheocrenic’ etc. should be used as adjectives.

Reply: We have controlled and changed it in the entire manuscript.

• Line 52: “continuous gradient”: not always… there are many natural discontinuities, and springs in particular can be very isolated.

Reply: We have added “generally”, allowing other possibilities.

• The language is already good but needs some editing, in particular in important parts of the text such as the Abstract and the Discussion. In an attempt to help to accomplish this, I provide the following edits.

Reply: All the sentences were modified according to reviewer’s suggestions, except:

1. Lines 436-437: Awkward sentence: Please re-write.

There was changed the entire paragraph

2. Line 473: Change to “flows there are many”.

If in the watersheds of these flows were many spring wetlands…

---

## [Editor Report · Decision Letter 1]

9 Mar 2020

Helocrenic springs as sources of nutrient rich fine particulate organic matter in small foothill watershed

PONE-D-19-31487R1

Dear Dr. Tichá,

We are pleased to inform you that your manuscript has been judged scientifically suitable for publication and will be formally accepted for publication once it complies with all outstanding technical requirements.

With kind regards,

Kai Yue, Ph.D.

Academic Editor

PLOS ONE

Additional Editor Comments (optional):

I am happy with the revisions, and think the current version is good for publication now.
---

## [Editor Report · Acceptance letter]

31 Mar 2020

PONE-D-19-31487R1 

Helocrenic springs as sources of nutrient rich fine particulate organic matter in small foothill watershed 

Dear Dr. Tichá:

I am pleased to inform you that your manuscript has been deemed suitable for publication in PLOS ONE. Congratulations! Your manuscript is now with our production department. 

With kind regards,

on behalf of

Prof. Kai Yue 

Academic Editor

PLOS ONE